# Alterations of Amino Acid Concentrations and Photosynthetic Indices in Light Irradiated *Arabidopsis thaliana* during Phytoextraction

**Yanmei Chen** [1], **Longfei Liang** [2], **Dan Chen** [1], **Tian Gan** [3], **Min Cao** [4] and **Jie Luo** [1,*]

1    College of Resources and Environment, Yangtze University, Wuhan 430100, China;
     ymchen831219@yangtzeu.edu.cn (Y.C.); Dan630v@163.com (D.C.)
2    Geophysical Exploration Brigade of Hubei Geological Bureau, Wuhan 430056, China; wtdllf@sina.com
3    School of Civil Engineering, Shandong University, Jinan 250100, China; jay-so.miss@foxmail.com
4    College of Natural Sciences, University of Leicester, University Road, Leicester LE1 7RH, UK;
     elover_min@hotmail.com
*    Correspondence: gchero1216@hotmail.com

**Abstract:** Hyperaccumulation of heavy metals is substantial in some hyperaccumulators; however, few studies have been conducted to reveal the effect of light irradiation on the variations of representative amino acids and photosynthetic indices, which can represent the antioxidant capacity of plants during phytoremediation. The physiochemical responses of *Arabidopsis thaliana* to Cd stress were compared during six light irradiation treatments. The results of the experiment showed that the stress adaptation of *A. thaliana* was enhanced in all light irradiation treatments, except for monochromatic blue-light irradiation. The concentrations of glutamic acid and glutamine decreased significantly in pure blue light compared with the other treatments. The decrease in the concentrations of these two amino acids might be induced by an intensive biosynthesis of defensive factors, as manifested in the declined photosynthetic indices. Concentrations of aspartic acid and asparagine involved in the ammonification, absorption, and transportation of nitrogen in vascular plants increased in the red and blue combined irradiation treatments compared with the control, corresponding to the improved photosynthetic capacity of the species. The concentration of proline, which can represent environmental stresses including metal toxicity and excessive light energy, generally increased with an increasing ratio of blue light. This study proposes the key roles of amino acids and photosynthetic indices in light-motivated Cd pollution responses in *A. thaliana*.

**Keywords:** phytoextraction; *Arabidopsis thaliana*; amino acid; photosynthetic index; light irradiation

## 1. Introduction

With the rapid development of the economy and society, environmental contamination has become increasingly severe. Soil pollution has gained worldwide concern because soil is an important non-renewable resource that provides habitats for terrestrial creatures and a medium for crop cultivation. Cd is regarded as one of the most toxic heavy metals to plants and animals owing to its high bioavailability and toxicity [1]. Approximately 30 million kilogram of Cd is discharged into environmental media annually, with an estimated 13 million kilograms released from anthropogenic activities [2]. Electronic waste (e-waste) recycling is one of the most environmentally harmful anthropogenic activities. The contaminated soils treated in the present study were sampled from an e-waste disassembling and recycling town located in southern China. A large area of soils in the town has been polluted by disassembling activities owing to primitive dismantling methods [3]. Environmentally harmful businesses have deleterious effects on the health of the local inhabitants, especially infants and children. For example, Cai et al. (2019) [4] found that metal exposure in Guiyu resulted in an increase in child sensory integration difficulties, compared with a nearby town not involved in the disassembling business. Therefore, it

is necessary to develop an environmentally friendly and economically viable method for decontaminating Cd-polluted fields, especially agricultural land.

The use of green plants, including hyperaccumulators and high-biomass-generating plants to stabilize or remove pollutants in soils, a technology called phytoremediation, is a developing feasible method for in situ decontamination [5,6]. Plants are used to extract metals from substrates and migrate them to aerial parts (phytoextraction) or to decrease their activity and toxicity in soils (phytostabilization). Both phytoextraction and phytostabilization should allow plants to accumulate as many metals as they can improve the remediation effect.

When Cd enters plant cells, it can cause changes in numerous biochemical processes at the cellular and subcellular levels by decreasing enzymatic activities, disturbing the functional groups of metabolites, replacing nutrients, and damaging membrane integrity [7]. Damaged cells cannot perform their physiological functions normally, thereby obstructing the conducting system and diminishing photosynthetic efficiency, resulting in considerable deleterious effects on the health of plant tissues [8,9].

In response to stress induced by excessive Cd, plants have developed multiple adaptation mechanisms to protect cells, such as activating signal transduction processes and biosynthesizing more proteins and pigments with antioxidant capacity in the impaired organ [5,10]. However, energy consumption processes require higher photosynthetic efficiency and perturb the normal physiological functions of the plants [11]. As precursors for protein biosynthesis, amino acids play a key role in the development and metabolic processes of plants. It has been reported that plants that accumulate large quantities of heavy metals are associated with a substantial accumulation of free amino acids [12]. In addition, amino acids are involved in the biosynthesis of phytochelatin, which can decrease the biological activity of metals by forming stable metal–chelate complexes [13].

Photosynthetic indices have been proven to be especially sensitive to Cd, which can decrease the photosynthetic efficiency of the photosynthetic apparatus by disturbing electron transportation, breaking $CO_2$ fixation, and resulting in stomatal closure [14]. Therefore, observations of photosynthetic indices are a nondestructive and sensitive technology for estimating the health conditions of plants [8,15]. Furthermore, the weakening of the photosynthetic efficiency leads to a decline in the replenishment of metabolic energy for N absorption and ammonification, disturbing the synthesis of amino acids [16]; thus, the relationship between amino acids and photosynthetic indices should be revealed during phytoremediation.

Light is an essential natural resource for modulating the germination, sprouting, flowering, and bearing fruit of plants. At the molecular level, light acts as an energy source and is involved in the biosynthesis of chlorophyll, carotenoids, and amino acids [17]. In addition, as a signaling source, light can regulate apical dominance through cryptochrome and phytochrome, respectively [18]. At the macro level, the utilization of light in agronomy and horticulture has become increasingly popular because it is a physical trigger in plants rather than a chemical trigger. Alterations in the morphology and function of plant tissues have been reported owing to changes in light quantity, quality, and duration, and these alterations have been shown to be species-specific. For instance, Li et al. (2010) [19] suggested that blue light significantly increased the chlorophyll level, leaf thickness, palisade parenchyma length, and stomatal area of *Gossypium hirsutum* compared with the control, while Zou et al. (2020) [20] reported that blue light decreased the growth rate, root length, and photosynthetic efficiency of *A. thaliana*. Our previous work also found that a proper blue/red light ratio can increase the Cd removal efficiency of *Noccaea caerulescens* by simultaneously enhancing its metal uptake capacity and oxidation resistance [21].

*Arabidopsis thaliana* has become a popular plant for the analysis of metal accumulation and decontamination [22,23]. However, reports on the roles and interrelation of amino acids and photosynthetic indices in the phytoremediation of Cd by *A. thaliana* under different light irradiation treatments are limited. It is safe to hypothesize that different light irradiation treatments would result in different physiological responses of plants to

environmental stresses, including Cd pollution. The current study aims to estimate the influences of different light treatments on biomass yield and Cd extraction capacity of *A. thaliana*, assess variations in the accumulation of amino acids and photosynthetic indices in the species under different light treatments, and reveal the resistance of the species to Cd stress by focusing on its micro indices.

## 2. Materials and Methods

### 2.1. Soil and Plant

Seeds of *A. thaliana* (wild-type) were sampled from a metalliferous field and cultivated according to the process conducted by Szopiński et al. (2019) [24]. Briefly, after being sterilized using 50% ethanol, the seeds were germinated in disinfected pots filled with vermiculite after further light irradiation treatments.

Considering the geochemical background and topography of the study region, 200 top-soil samples ($0-20$ cm) were collected. The air-dried and sieved (2-mm mesh) samples were blended to obtain a composite substrate. The blended substrate was equilibrated by four saturating and drying processes to obtain a homogeneous distribution of the metal. After equilibration, the substrate was divided into approximately 6000 g aliquots and filled in disinfected pots for *A. thaliana* transplantation.

### 2.2. Light Irradiation Treatment

After 3 weeks of acclimation, 10 seedlings were transplanted to each pot and thinned to five before light illumination. Six light irradiation treatments with five replications, including monochromatic blue ($B_{100}$), 25% red and 75% blue ($B_{75}$), 50% red and 50% blue ($B_{50}$), 90% red and 10% blue ($B_{10}$), and monochromatic red ($B_0$), as well as an incandescent lamp (control), were conducted under controlled conditions (22 °C, 60% relative humidity). The illumination intensities were set at 200 μmol·m$^{-2}$·s$^{-1}$ photosynthetic photon flux density at an 8 h·night/16 h·day photoperiod, and a spectrometer was used to monitor the constancy of light illumination. Four weeks after light irradiation, all seedlings were harvested and divided into roots and shoots. The plant tissues were cleaned by running water to remove foreign materials and soaked in 10 mM EDTA to eliminate the adsorbed ions.

### 2.3. Cd Analysis and Quality Control

For the analysis of Cd, dried soil and plant samples were pulverized and sieved by 74-μm meshes. The powders were digested using aqua regia at 120 °C for 6 h and then diluted with deionized water [25]. After cooling, the digestion was filtered via a 0.45-μm membrane, and the concentration of Cd in the filtrate was quantified by ICP-MS (Agilent 7700, Santa Clara, CA, USA). An Agilent multi-element calibration standard (Agilent, ZCA-8500-6940) was applied to each measurement batch for quality control. The reference material GBW10010 (plant) was also analyzed for quality assurance.

The content of soil Cd ($3.83 \pm 0.46$ mg kg$^{-1}$) was only analyzed at the beginning of the treatment because the duration of this experiment was not long enough for *A. thaliana* to change the pseudototal Cd content. The phytoremediation efficiency of the species was evaluated based on its biomass generation and Cd uptake capacity.

### 2.4. Measurement of Chlorophyll Fluorescence

A fluorometer (PAM 2100, WALZ Corporation, Forchheim, Germany) was used to measure the chlorophyll fluorescence. Minimal ($F_0$) and maximal ($F_M$) fluorescence were measured using a saturating flash after a 30 min dark adaptation, and then the plant leaves were illuminated using an actinic light for $F'$ (steady state fluorescence level in light-adapted tissues) and $F'_M$ (maximal chlorophyll fluorescence in light-adapted tissues) measurements according to the process performed by Li et al. (2015) [26]. The chlorophyll fluorescence parameters, including maximum quantum yield of photosystem II (PSII) after dark adaptation of ($F_v/F_M$), effective quantum yield of PSII ($Y(II)$), photochemical

quenching (qP), and non-photochemical quenching (NPQ) were calculated according to Agrawal et al. (2016) [27].

### 2.5. Measurement of MDA, $H_2O_2$

MDA was determined according to the process described by Zhang et al. (2019) [28]. Briefly, the plant leaves (0.5 g) were homogenized in an 8 mL thiobarbituric acid (TBA, 10%) solution. After centrifugation at $4000 \times g$ for 20 min, the mixture of the filtrate (2 mL) and TBA (2 mL, 0.6%) was boiled for 30 min. After cooling, the absorbance of the solution was determined at 532, 450, and 600 nm to calculate the MDA content using the following formula:

$$MDA = 6.45 \times (A_{532} - A_{600}) - 0.56 \times A_{450}$$

Fresh leaves (0.1 g) were homogenized in acetone (1 mL), and then a hydrogen peroxide assay kit (BC3595, Solarbio, Beijing, China) was used to determine the concentration of $H_2O_2$ in the plant leaves. The measurements were performed according to the manufacturer's instructions.

### 2.6. Measurement of Amino Acids

The plant tissues were pretreated according to the method suggested by Xu and Xiao (2016) [29] for chromatographic analysis. In brief, 0.15 g of the plant samples was shaken in 1.8 mL trifluoroacetic acid (10% v/v) for 5 min. After ultrasonic treatment for 5 min, the solution was centrifuged at 12,000 rpm for 15 min. After filtration, the residue was extracted using the same procedure, and the recovered supernatants were combined to improve the precision of the analysis. The combined solution was filtered by a 0.2 μm polycarbonate membrane (Whatman, Maidstone, UK) and then analyzed using a high-performance liquid chromatograph (HPLC, Agilent 1200, Santa Clara, CA, USA). Five amino acids, including glutamic acid (Glu), glutamine (Gln), aspartic acid (Asp), asparagine (Asn), and proline (Pro), were measured.

### 2.7. Statistical Analysis

All the data in this study are shown as the average values of five replicates. The influence of light irradiation treatments on the dry weight, Cd concentration, chlorophyll fluorescence parameters, pigment levels, and amino acid levels were determined using one-way analysis of variance. Comparisons of means were performed by Fisher's LSD post-hoc tests at 0.05 probability. The statistical analysis of data was executed using SPSS 15.0.

## 3. Results and Discussion

### 3.1. Plant Growth

The dry weight of *A. thaliana* in the current experiment ranged from 0.40 to 1.32 g, which was slightly lower than the values recorded in previous studies [30,31]. Light irradiation significantly enhanced the biomass of the plant roots compared with the control, except for monochromatic blue light, which resulted in the lowest value at the termination of the treatment (Figure 1). There was a proportional decrease in the dry weight of the plant roots with increasing ratios of blue light, except for $B_{10}$, which triggered the highest value. All the light treatments enhanced the biomass yield of the above ground parts of *A. thaliana* compared with the control during the experiment, and the highest value was observed in B10. As shown in Figure 1, $B_0$ had a lower aerial part to whole plant ratio than the control, while other treatments had higher ratios. In general, the dry weight of the aerial parts increased with an increase in the ratio of blue light until it reached a peak of 1.03 g in $B_{50}$.

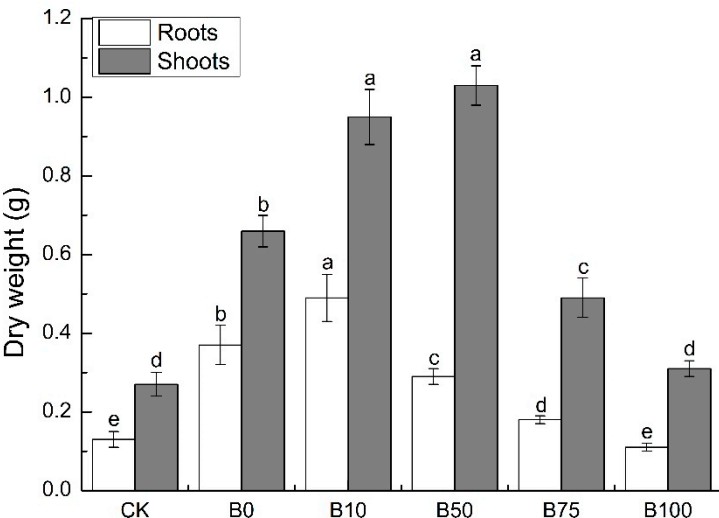

**Figure 1.** Dry weight of *A. thaliana* under different light treatments.

### 3.2. Cd Content

As a Cd accumulator, *A. thaliana* can accumulate significantly higher levels of the metal in its aerial parts (generally higher than 100 mg·kg$^{-1}$) relative to the rhizosphere soil [32]. Compared with previous treatments that grew the species in a sewage sludge applied soil [31], or a Cd-spiked substrate [33], in the current experiment, *A. thaliana* in the control had significantly lower Cd levels in its aerial parts, which did not achieve the criterion of 100 mg·kg$^{-1}$ for Cd.

Light irradiation treatments changed the Cd content in the plant roots and shoots. There was a decline in the Cd concentrations in the plant roots with increasing doses of blue light. Relative to the control, all treatments increased the content of Cd in the plant shoots, except for $B_0$, which declined the value significantly. The concentrations of Cd in the plant shoots elevated with enhancing ratios of blue light until it achieved a peak of 121.6 mg·kg$^{-1}$ in $B_{50}$, after which the metal in the aerial parts declined significantly (Figure 2). Notably, in $B_{10}$, $B_{50}$, and $B_{75}$, the concentrations of Cd in the aerial parts of *A. thaliana* met the criterion of a Cd hyperaccumulator.

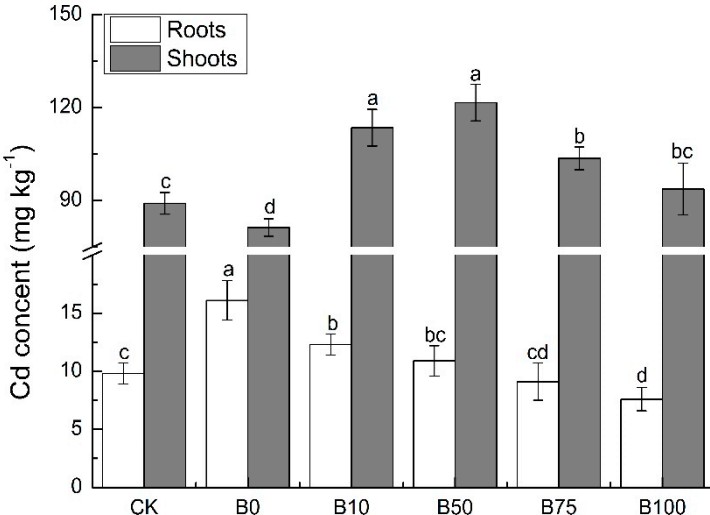

**Figure 2.** Content of Cd in *A. thaliana* roots and shoots under different treatments.

The translocation factor (TF), calculated by dividing the content of Cd in the aerial parts by that in the roots, was used to assess the Cd migration and accumulation ability of

the species [34]. The TF values in the control, $B_0$, $B_{10}$, $B_{50}$, $B_{75}$, and $B_{100}$ were 9.1, 5.0, 9.2, 11.2, 11.4, and 12.3, respectively. The results indicated that monochromatic red light drove *A. thaliana* to translocate more Cd into its roots, while blue light had the opposite effect.

### 3.3. MDA and $H_2O_2$ Content

Both heavy metals and excessive illumination can induce the generation and accumulation of superoxide radicals [35,36]. In the present study, the accumulation of $H_2O_2$ in the plant tissues was measured to estimate the level of oxidative damage under various treatments. As shown in Figure 3, the levels of $H_2O_2$ in the above ground parts of *A. thaliana* elevated with enhancing ratios of blue light. Relative to the control, three red–blue combinations ($B_{10}$, $B_{50}$, and $B_{75}$) increased the concentrations of $H_2O_2$ slightly ($p > 0.05$), while monochromatic red and blue light caused a decrease and increase, respectively. The accumulation of $H_2O_2$ did not correlate well with the variation in Cd content ($r = 0.5$, $p = 0.3$). For instance, the highest $H_2O_2$ was recorded in $B_{100}$, in which the content of Cd in plant shoots was similar to that of the control. These results indicate that Cd accumulation might not be a unique mechanism for the increased $H_2O_2$ concentration observed in this study.

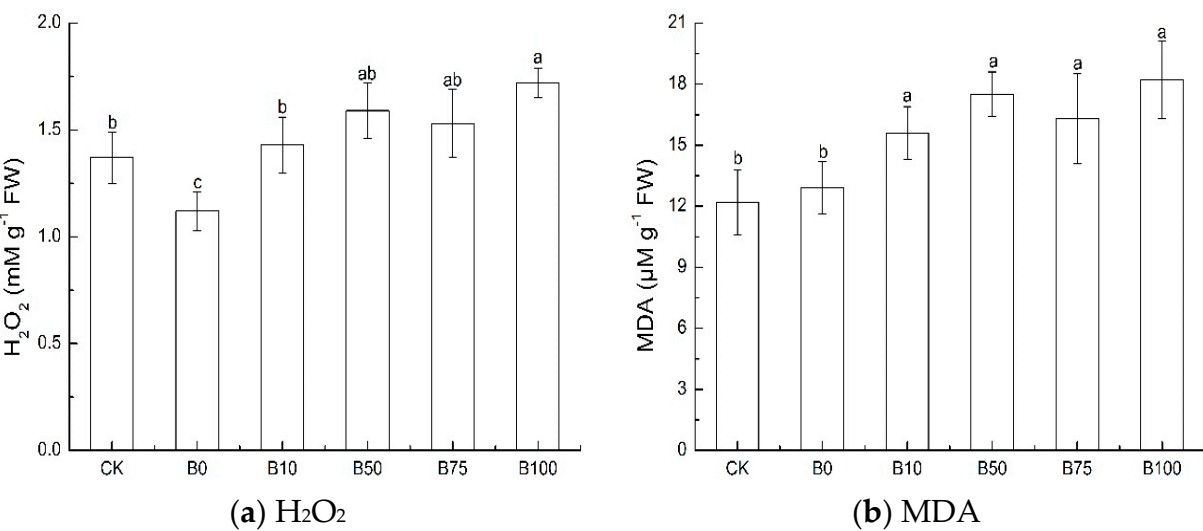

**Figure 3.** Levels of $H_2O_2$ and MDA in *A. thaliana* shoots under different treatments.

MDA is regarded as an effective indicator of lipid peroxidation [37]. Except for $B_{100}$, which had the highest MDA content, the levels of MDA in the plant shoots elevated with enhancing ratios of blue light until it achieved a peak at $B_{50}$ (Figure 3). Relative to the control, all treatments increased the level of MDA in the above ground parts of *A. thaliana*, except for monochromatic red light. The positive correlation between MDA and Cd content was also not significant ($r = 0.6$, $p = 0.2$).

### 3.4. Chlorophyll Fluorescence Characteristic

Chlorophyll fluorescence parameters can reflect the status and properties of photosynthetic processes with variations in light assimilation, redistribution, divergence, and conversion [38].

As shown in Figure 4, $F_v/F_M$, which is regarded as a dependable indicator of photoinhibition, showed no significant difference in all treatments. However, a clear trend that $F_v/F_M$ elevated with enhancing doses of blue light until it achieved a peak at $B_{75}$, after which the lowest value among all treatments was observed in $B_{100}$. A decrease in $F_v/F_M$ implies a decrease in the maximum quantum yield of an open photosynthetic system II (PSII), as well as an increase in light dissipation, indicating damage to the reaction center [39]. All the light treatments, except $B_{100}$, had no significant effect on $Y(II)$. The combination of red and blue light increased $Y(II)$ in a variation from 3.0% to 14.5%, and the monochromatic blue light

decreased $Y(II)$ significantly by 33.1% compared with the control. The recorded decrease in $Y(II)$ indicates that monochromatic blue light aggravated the light energy dissipation of the antenna pigment and resulted in a decline in the efficiency of excited energy trapped by PSII [40].

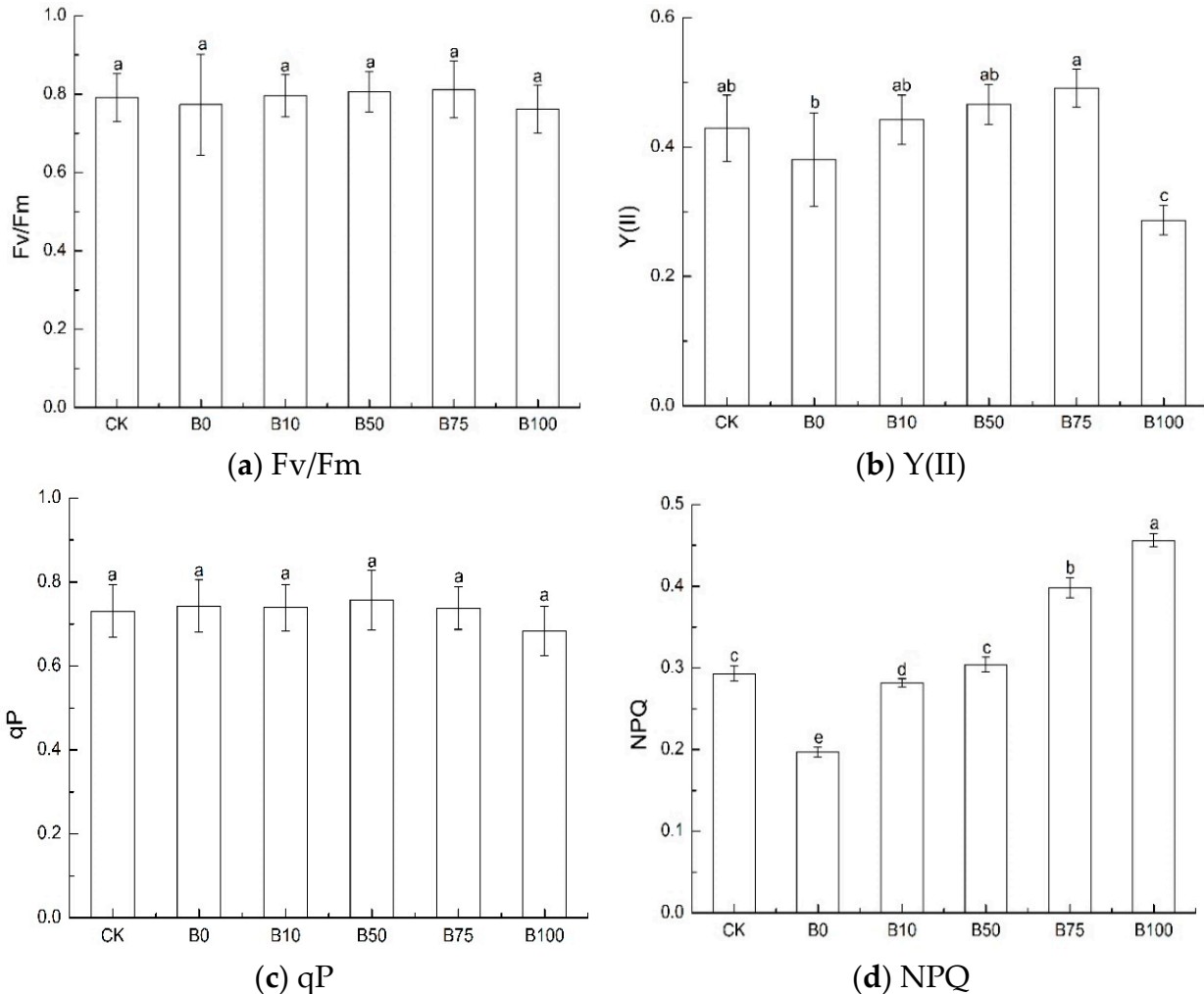

**Figure 4.** Chlorophyll fluorescence parameters of *A. thaliana* under different treatments.

All the light irradiation treatments had no significant effect on qP, and the lowest qP was observed in $B_{100}$. NPQ gradually decreased in contrast to the increase in qP, and monochromatic blue light increased NPQ by 57.8%, 131.5%, 61.7%, 50.0%, and 14.6% relative to the control, $B_0$, $B_{10}$, $B_{50}$, and $B_{75}$, respectively. Considerable amounts of the captured light are used to facilitate photosynthesis (qP) or are dissipated as nonradiative heat (NPQ). It is a protective mechanism of the photosynthetic system that may minimize the oxidative damage induced by excessive illumination [35,41].

### 3.5. Amino Acid

The results of the experiment suggested different influences of light irradiation on the amino acid concentrations. The concentration of Glu in the aerial parts of *A. thaliana* increased slightly in $B_0$, but other treatments significantly decreased the concentration (Table 1). Except for monochromatic red, all the light irradiation treatments decreased the content of Gln compared with the control.

**Table 1.** Levels of amino acids ($\mu$mol kg$^{-1}$ FW) in *A. thaliana* under different treatments.

| | Glu | Gln | Asp | Asn | Pro |
|---|---|---|---|---|---|
| CK | 973 ± 11 a | 2331 ± 161 b | 1012 ± 89 cd | 3129 ± 73 c | 363 ± 12 d |
| B0 | 981 ± 31 a | 2591 ± 87 a | 1087 ± 76 c | 3256 ± 161 bc | 337 ± 13 e |
| B10 | 902 ± 20 b | 2234 ± 107 b | 1352 ± 119 ab | 3579 ± 113 a | 387 ± 11 c |
| B50 | 863 ± 19 c | 1981 ± 126 c | 1434 ± 56 a | 3432 ± 95 ab | 412 ± 9 b |
| B75 | 817 ± 17 d | 1862 ± 89 c | 1223 ± 89 bc | 3316 ± 63 b | 391 ± 7 c |
| B100 | 726 ± 29 e | 1573 ± 102 d | 932 ± 51 d | 2671 ± 89 d | 439 ± 13 a |

Different letters mean significant differences in amino acid concentrations in *A. thaliana* shoots ($p < 0.05$) judged by Fisher's LSD post-hoc tests.

Asp and Asn, which are formed by Gln and Glu, translocate organic N from the source to sink in plants. Our results demonstrated that the content of Asp raised with the enhancing ratio of blue light until it achieved a peak at B$_{50}$, after which the concentration decreased significantly (Table 1). The variation trend of Asn was similar to that of Asp, but the highest content was observed in B$_{10}$.

Pro is a reliable diagnostic indicator of different abiotic stresses, including Cd toxicity [42] and light illumination [43]. In the current study, the levels of Pro were significantly higher in B$_{50}$, which had the highest Cd content in the aerial parts of *A. thaliana*, and B$_{100}$, which accepted the monochromatic blue light.

### 3.6. Phytoremediation Efficiency

The difference in the initial soil Cd concentration and the acceptable threshold (0.3 mg·kg$^{-1}$) multiplied by the weight of the soil (6 kg per pot) was the amount of the metal that needed to be removed. The Cd accumulation capacity (AC) of *A. thaliana* was calculated as the product of its dry weight and Cd content. The necessary growing cycles to decontaminate the soil were calculated as the quotient of the excess Cd and the AC divided by five. According to the biomass generation and Cd accumulation capacity of *A. thaliana*, the AC of the species were 0.13, 0.30, 0.57, 0.64, 0.26, and 0.15 mg Cd per pot in the control, B$_0$, B$_{10}$, B$_{50}$, B$_{75}$, and B$_{100}$, respectively. Therefore, 168, 72, 38, 33, 80, and 142 growing cycles are required for the species to clean the soil under the corresponding treatments. Obviously, B50, which reduced the required time by 80% compared with the control, was the optimal strategy.

As shown in Figure 1, the highest dry weight of *A. thaliana* was not observed in B$_{50}$, but the relatively higher Cd extraction ability of the species counteracted the reduction in biomass. However, both the biomass yield and metal extraction ability of *A. thaliana* decreased with the increasing proportion of blue light, reducing the phytoremediation efficiency of the species.

## 4. Discussion

### 4.1. Responses of Biomass Yield and Cd Accumulation to Light Irradiation

The responses of plant species to light illumination are specific to species. Lin et al. (2013) [44] found that monochromatic red LED increased the fresh and dry weight of *Lactuca sativa* significantly when compared with the control, in agreement with the results of our study. In contrast, our previous study suggested that red light increased the root dry weight of *Noccaea caerulescens*, but decreased its leaf biomass, and the improved root biomass could not offset the reduction in the leaf biomass [21]. Interestingly, the present study found that although monochromatic blue light resulted in the lowest dry weight of roots and aerial parts of *A. thaliana*, the application of a small quantity of blue light can significantly enhance the biomass generation capacity of the species, as manifested in the highest biomass of *A. thaliana* in B$_{10}$. This finding agrees with the reports of Hogewoning et al. (2010) [45] who found that only 7% blue light was sufficient to prevent any overt dysfunctional photosynthesis of *Cucumis sativus*.

To reveal the effect of illumination on plant health at the cellular level, Dong et al. (2014) [46] reported that a single light might induce a burst of reactive oxygen, resulting in

a surge in superoxide radicals, which would induce oxidative damage to cell membranes. However, in the presence of heavy metals, the physiological responses of plants will become more complex. Suitable light irradiation can cause plants to accumulate more metals, which would have deleterious effects on plant cells [47]; therefore, the variation in the Cd concentration in the plant tissues under different light treatments was examined.

### 4.2. Influences of Light Irradiation on Cd Accumulation

The primary toxic mechanisms of Cd on plant cells are the reduction of glutathione, inhibition of protein synthesis, and substitution of pivotal elements, including Zn and Mg, in the chloroplasts [48]. Therefore, more energy is consumed to counteract the detrimental effects caused by Cd, resulting in the redistribution of nutrients and disturbing the normal function of the plant organs. Light irradiation treatments generally increased Cd content in the roots and the above ground parts of *A. thaliana*, but there was a weak positive correlation between the Cd content and plant dry weight (r = 0.7, $p$ = 0.08), indicating that a suitable light treatment might alleviate the negative effects of Cd on plant health.

The impacts of light quantity, quality, and duration on nutrient cycling in plant tissues have been fully reported, but the uptake and translocation of pollutants including Cd in the soil and plant under the impact of illumination have rarely been revealed. To date, only Kwon et al. (2017) [49] have studied the influence of light irradiation (red, blue, yellow, and their combination) on the adsorption and absorption of Cu and Zn by microalgae and reported that microalgae grown under monochromatic red light had the highest Cu and Zn decontamination efficiency. However, the physiological response of algae to light might differ from that of plants. It has been suggested that the transpiration rate of plants can influence the metal uptake capacity of plants. For instance, Wan et al. (2015) [50] evaluated the role of transpiration in the extraction of As from the perspective of transpiration modulation and found that *Pteris vittata* grown in a humid environment had a 40% higher transpiration rate and 40% higher leaf As level compared with plants grown in dry habitats. Light irradiation can influence the transpiration rate of plants by affecting their photosynthetic system; therefore, the influences of light treatment on oxidative damage photosynthetic indices were discussed.

### 4.3. Relationship among Oxidative Damage, Photosynthetic Indices, and Amino Acid Content

It has been reported that the accumulation of pollutants, including metals, in plant tissues results in oxidative damage by promoting the generation of reactive oxygen species, which have deleterious effects on the cell membrane [51–53]. However, the increase in $H_2O_2$ and MDA was not significantly related to the extraction and accumulation of the metal in *A. thaliana*. In general, the pollutant uptake and biomass generation contradict each other. The contradictory results found in this study could be supported by Wu et al. (2018) [54], who reported that the biomass of *Malva crispa*, *Celosia argentea*, and *Celosia cristata* did not decline as the Cd content increased, and it was even higher than that of the control. They attributed low-dose promotion to the hormetic dose responses. A minute dose of Cd can increase the growth rate of some plants by activating antioxidant enzymes and hormones in the plant tissues. In addition, light illumination itself can impact the health of plants by modulating the electron transport rate and photochemical quenching of the photosystems. Therefore, variations in amino acid and chlorophyll fluorescence parameters are discussed. It is worth noting that cell death in *A. thaliana* can be regulated by apoplastic ROS and jasmonic acid [55], and cell death in the species under different light treatments should be revealed in future.

Combining the result that a relatively low Cd content in the aerial parts of *A. thaliana* was recorded in $B_{100}$, the decreased qP and increased NPQ indicated that monochromatic blue light induced photo-oxidative damage to the photosynthetic apparatus, and a defense mechanism was activated to alleviate the damage. This finding is inconsistent with the results of Li et al. (2017) [56], who found that qP and NPQ of wheat under ultraviolet light (UV-B) stress declined significantly compared with the control. This might be because the

oversupplied energy provided by ultraviolet light can damage the self-protecting ability of the PSII reaction centers because their maximum capacity is overwhelmed. This suggests that monochromatic blue light does not overwhelm the self-protecting ability of PSII in *A. thaliana*.

Considering the result of MDA in $H_2O_2$ concentrations in plant tissues, the oxidative damage in *A. thaliana* was mainly induced by monochromatic blue light irradiation, rather than the accumulated Cd. In contrast, in $B_{10}$, $B_{50}$, and $B_{75}$, oxidative damage may have been induced by the concentration of Cd.

The reduction in the Glu content might be induced by intensive biosynthesis of defense elicitors, which require more energy. Gln, which can represent the N utilization pathway, is one of the dominant amino acids in *A. thaliana*. Gln is a donor catalyzed by glutamate synthase, which is a key enzyme in the assimilation of inorganic N [57]. This process represents the energy status of the photosynthetic apparatus. Aspartate aminotransferase controls the biosynthesis of Asp, and its activity is sensitive to environmental stresses [58]. The significantly greater reduction in the Asp concentration in $B_{100}$ indicates increased stress under the monochromatic blue light treatment. Both the Cd accumulation and the increasing ratio of blue light (when higher than 50%) induced negative effects on the synthesis of Asn. Considering the variation trends of amino acids associated with the accumulation of Cd and light irradiation treatments, we hypothesized that monochromatic blue light could induce greater damage to the photosynthetic system of *A. thaliana* than Cd. This hypothesis should be verified through more sophisticated experiments.

## 5. Conclusions

The results of this experiment demonstrate that the phytoremediation efficiency of *A. thaliana* could benefit from combinations of blue and red light, especially $B_{10}$ and $B_{50}$, which resulted in better biomass generation and Cd accumulation capacity. Monochromatic blue light induced significantly lower biomass and Cd content of *A. thaliana* compared with other light irradiation treatments, although it enhanced the phytoextraction ability of the species relative to the control. As a high-energy light, blue light induced photoinhibition in the plant leaves, as manifested by the decreased photosynthetic indices, including Fv/FM, Y(II), and qP in $B_{100}$. However, the significantly higher NPQ in $B_{100}$ compared with other light irradiation treatments indicates that the photosynthetic system can dissipate excessive light energy as heat to alleviate continuous oxidative damage. In other words, blue light did not overwhelm the self-protecting ability of the species. The variations in amino acid concentrations suggest that both Cd accumulation and a high ratio of blue light could trigger deleterious effects on *A. thaliana*, thus decreasing its phytoremediation efficiency. In addition, the continuous depletion of stratospheric $O_3$ could result in an increase in ultraviolet light, which contains significantly more energy compared with blue light in the near surface, and the effects of ultraviolet light on the phytoremediation efficiency of plants should be studied in future research. Cell death and other ROS including superoxide and hydroxyl radicals, which can also reflect the physiological responses of plants to external environments will be measured in our future work.

**Author Contributions:** Conceptualization, J.L.; methodology, J.L. and Y.C.; software, D.C. and T.G.; validation, Y.C. and L.L.; formal analysis, J.L. and D.C.; investigation, L.L. and T.G.; resources, J.L. and Y.C.; data curation, M.C.; writing—original draft preparation, Y.C. and J.L.; writing—review and editing, M.C. and T.G.; visualization, Y.C. and M.C.; supervision, J.L.; project administration, J.L.; funding acquisition, J.L. All authors have read and agreed to the published version of the manuscript.

**Funding:** This research was funded by the National Natural Science Foundation of China, grant number 21876014 and the Natural Science Foundation of Hubei Province of China, grant number 2018CFB258.

**Institutional Review Board Statement:** Not applicable.

**Informed Consent Statement:** Not applicable.

**Data Availability Statement:** Not applicable.

**Conflicts of Interest:** The authors declare no conflict of interest.

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
