# Peer review of "Alterations of Amino Acid Concentrations and Photosynthetic Indices in Light Irradiated Arabidopsis thaliana during Phytoextraction"

_sustainability, doi:10.3390/su13147720_

Round 1

Reviewer 1 Report

Its well-written manuscript and results could largely benefit the readers and the community.

however, here are my modifications for enhancing the current quality.

figure 1: Can the authors show Fresh weight in parallel? also, accompanying photographs can be helpful

can authors accompany their H2O2 with DAB staining?

Can measure other ROS species at least by immunostaining?

can authors show that they are not measuring cell death? by staining assays.

minor, 

please abbreviate Cd?

Reviewer 2 Report

The authors propose a manuscript titled: Alterations of amino acid concentrations and photosynthetic indices in light irradiated Arabidopsis thaliana during phytoex traction.

The manuscript are well written and argued.

The topic is closely falls within the aims and scope of the journal. 

Material and methods not need  modifications.

The Results and discussion of focus on the main points while justification of the findings are well supported by references.

The data provided are sufficient and the statistical analysis of the results is well presented.  

Figures and Table clearly present the data.

Reviewer 3 Report

The manuscript entitled “Alterations of amino acid concentrations and photosynthetic indices in light irradiated Arabidopsis thaliana during phytoextraction by Chen et al., is written well. I will suggest few minor revision before its acceptance

Introduction

Please check the reference style of MDPI and adjust according to journal

Line 85, please add the word “light” after “blue/red”

Please describe the hypothesis in the last paragraph to achieve the mentioned objective

Materials and Methods

Please move the descriptive paragraphs to the introduction part

Statistical analysis

The statement “Differences were regarded significant at P < 0.05. SPSS 15.0 was used to perform the statistical analyses” is confusing. What are the meanings of this statement? Usually, we do data analysis to check the significance. Isn’t it?

How was the lettering done? Is it through LSD?

Results and Discussion

I will advise if result can be a separate part and discussion a part
